# The Type IV Pilus of Plasmid TP114 Displays Adhesins Conferring Conjugation Specificity and Is Important for DNA Transfer in the Mouse Gut Microbiota

Nancy Allard,[a] Kevin Neil,[a] Frédéric Grenier,[a] Sébastien Rodrigue[a]

[a]Département de biologie, Faculté des sciences, Université de Sherbrooke, Sherbrooke, Quebec, Canada

**ABSTRACT** Type IV pili (T4P) are common bacterial surface appendages involved in different biological processes such as adherence, motility, competence, pathogenesis, and conjugation. In this work, we describe the T4P of TP114, an IncI2 enterobacterial conjugative plasmid recently shown to disseminate at high rates in the mouse intestinal tract. This pilus is composed of the major PilS and minor PilV pilins that are both important for conjugation in broth and in the gut microbiota but not on a solid support. The PilV-coding sequence is part of a shufflon and can bear different C-terminal domains. The shufflon is a multiple DNA inversion system containing many DNA cassettes flanked by recombination sites that are recognized by a shufflon-specific tyrosine recombinase (shufflase) promoting the recombination between DNA segments. The different PilV variants act as adhesins that can modify the affinity for different recipient bacteria. Eight PilV variants were identified in TP114, including one that has not been described in other shufflons. All PilV variants allowed conjugative transfer with different recipient *Escherichia coli* strains. We conclude that the T4P carried by TP114 plays a major role in mating pair stabilization in broth as well as in the gut microbiota and that the shufflon acts as a biological switch modifying the conjugative host range specificity.

**IMPORTANCE** Conjugative plasmids are involved in horizontal gene transfer in the gut microbiota, which constitutes an important antibiotic resistance gene reservoir. However, the molecular mechanisms used by conjugative plasmids to select recipient bacteria and transfer at high rates in the mouse gut microbiota remain poorly characterized. We studied the type IV pilus carried by TP114 and demonstrated that the minor pilin PilV acts as an adhesin that can efficiently select target cells for conjugative transfer. Moreover, the *pilV* gene can be rapidly modified by a shufflon, hence modulating the nature of the recipient bacteria during conjugation. Our study highlights the role of mating pair stabilization for conjugation in broth as well as in the gut microbiome and explains how the host spectrum of a plasmid can be expanded simply by remodeling the PilV adhesin.

**KEYWORDS** conjugative plasmid, type IV pilus, shufflon, microbiota

Address correspondence to Sébastien Rodrigue, Sebastien.Rodrigue@USherbrooke.ca.

The authors declare a conflict of interest. Some aspects of the work presented in this study are part of patent application WO2020010452A1. All authors of the present manuscript except for F.G. are also co-authors of this provisional patent application. S.R. and K.N. have a financial interest in TATUM bioscience.

To thrive in their natural environment, bacteria have evolved diverse types of specialized appendages on their surfaces. These extracellular structures include type IV pili (T4P), which are complex proteinaceous assemblies involved in a wide variety of cellular functions, including cell motility (1, 2), pathogenicity (3, 4), adherence to biotic and abiotic substrates (5–8), DNA uptake (competence) (9), and DNA exchange (conjugation) (6, 10), and they can even act as nanowires carrying electric current (11). T4P shares core homologous components and similarities in terms of macromolecular architecture with the bacterial type II secretion systems (12), the competence pseudopilus, and the archaeal flagella (9, 13, 14). T4P have been observed in various Gram-

negative bacteria (4) such as *Pseudomonas aeruginosa* (1), *Neisseria*, *Vibrio cholerae* (15), and *Escherichia coli* (8, 12). Despite some structural differences, T4P are also found in many Gram-positive bacteria, including *Bacillus*, *Streptococcus*, and *Clostridium* (7, 13), in addition to several archaeal species (2).

While genes encoding T4P proteins are usually found within the chromosome of bacteria (9), they can also be located on mobile genetic elements such as conjugative plasmids. In that context, T4P generally contributes to the dissemination of mobile genetic elements in unstable environments where mobility, flow forces, and environmental factors could perturb bacterial interactions (6, 10, 16). Conjugation is a complex process that allows donor bacteria to transfer genetic material to a recipient cell through a sophisticated channel known as the type IV secretion system (T4SS) (17). The presence of a T4P or other extracellular structure, like a conjugative F-pilus, can facilitate contact with neighboring bacteria and mating pair formation by bringing the two cells close to each other through dynamic rounds of extension and retraction, presumably by depolymerization (18) or other unknown mechanisms (19). The T4P can play an important role in mating pair stabilization, allowing the deployment of the T4SS and subsequent transport of DNA (16, 17, 20).

The overall structure of the T4P consists of two main components, the extracellular pilus fiber and the cell envelope spanning complex (21). The latter comprises diverse assembly proteins, ATPase(s), and an inner membrane core protein, in addition to an outer membrane secretin channel in the case of Gram-negative bacteria (13). The pilus fiber is composed of very long polymers of fibrous proteins called pilins. Each polymer contains thousands of copies of the major pilin protein (PilS/A/E) in addition to a low abundance of a minor pilin subunit (PilV/E/W/X) that can act as an adhesin (9, 14, 22, 23). To enable their secretion, these structural subunits are synthesized as preproteins with a hydrophilic signal peptide of variable length, known as type III signal sequence (14), which is removed in the cytoplasm during filament assembly by a dedicated prepilin peptidase (24). While some minor pilins (e.g., CofB) have been determined to be located at the pilus tip (25–29), others are presumably incorporated sporadically along the pilus length (30). During assembly, the pilins are tightly packed into a helix in a way that the highly hydrophobic N-terminus is buried within the pilus core (14, 15). This results in an extremely thin pilus fiber ranging from 5 to 9 nm in diameter that can extend several micrometers from the cell body (3, 5, 17, 31).

T4P can be categorized into three different classes, termed types IVa, IVb, and IVc, based on the sequence of the major pilin subunit as well as differences in their assembly systems and functions (9, 12, 14, 32). Components of class IVa are generally scattered throughout the genome (2, 7, 33), and their assembly is relatively complex (21, 31). Conversely, type IVb and IVc pili comprise fewer biogenesis genes that are often grouped within the same operon (2, 31). All type IV major pilins share a conserved hydrophobic $\alpha$-helical N-terminus domain ($\sim$25 residues) that contains a characteristic type III signal sequence, almost always a glutamate residue at position 5 (4, 10), and a less conserved C-terminus either harboring a cysteine pair (4, 8, 15) or involved in hydrophobic interactions for stabilization (13, 14). Major pilins of class IVa are relatively homogenous in size ($\sim$150 to 175 residues), contain a short signal peptide ($\leq$10 residues), and most often have methylated phenylalanine at the N-terminus of the mature protein (14). In contrast, pilins of class IVb are more diverse, larger ($\sim$180 to 240 residues), and characterized by a longer signal peptide (13 to 30 residues) (3, 14, 34) with a variable N-terminal amino acid consisting of a methionine (methylated in some case), leucine, tryptophan, or serine (4). Members of the IVc family, also called Tad (tight adherence) pilus, Flp (fimbrial low-molecular-weight protein), or Fap (fibril-associated protein) (5, 9, 14), share a long signal peptide ($\leq$37 residues) but are considerably shorter in length ($\sim$50 to 80 residues). They also contain an Flp motif consisting of 20 hydrophobic residues at the N-terminus of the mature pilin, with adjacent glutamate and tyrosine residues at its center (5).

The T4P of conjugative plasmids is often associated with a shufflon (22). This multiple DNA inversion system has been identified in certain phages (35), as well as in

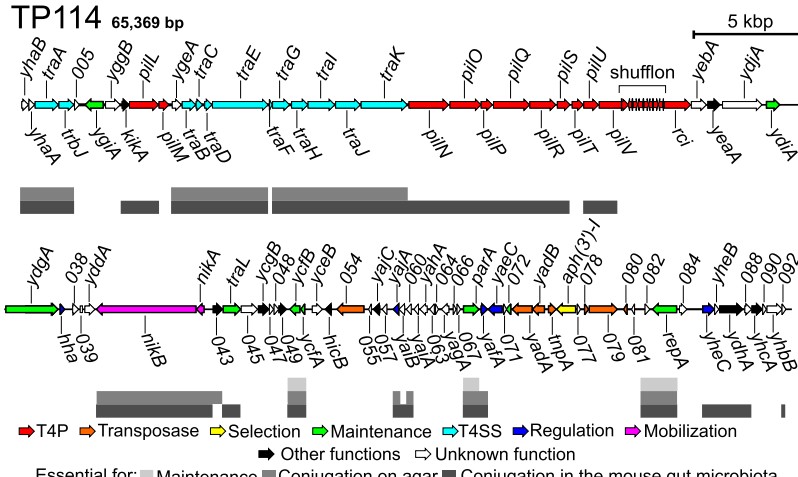

**FIG 1** Genetic map of conjugative plasmid TP114 (Kn^R) and gene essentiality status for plasmid maintenance and conjugation. Genes are colored-coded based on their predicted function. The position of the predicted shufflon is also indicated. Gene essentiality for plasmid maintenance, *in vitro* conjugation on agar, and *in vivo* conjugation in the mouse gut microbiota is shown in different tones of gray. Gene essentiality data were taken from Neil et al. (40).

several plasmids from different incompatibility groups (Inc) designated the I-complex plasmid group (including IncB/O, IncI1α, IncI1γ, IncI2, IncK, IncZ) (36) due to the morphological and serological properties of their pili. The shufflon consists of a *pilV* gene, including multiple DNA cassettes containing recombination sites and a shufflon-specific tyrosine recombinase of the integrase family, termed "shufflase." The shufflase, encoded by the *rci* (recombinase for clustered inversion) gene, can generate different variants of *pilV* by swapping the DNA cassettes comprised between two conserved recombination motifs (*sfx* sites), hence changing the *pilV* 3′ end. The *sfx* sites are composed of a conserved 7-bp sequence (core site), where DNA crossover occurs, with a highly conserved 12-bp right arm and a nonconserved 12-bp left arm. *rci*-mediated recombination between *sfx* sites that face each other causes inversion of the DNA segments independently or in groups (37). This biological switch introduces C-terminus variants of the PilV protein, allowing quick and reversible adaptation to environmental changes (36). The *pilV* variants encode different adhesins, all of which recognize specific structures on the surface of recipient cells during conjugation and allow mating in broth (6, 36, 38).

In this study, we characterized the T4P and shufflon of TP114, a conjugative plasmid from the IncI2 incompatibility group isolated from an *E. coli* strain in Scotland in 1967 (39) but recently shown to be required for high transfer rates in the mouse intestinal microbiota (40). We examined the T4P by fluorescence microscopy using maleimide-conjugated fluorophores (32). We also studied the TP114 shufflon and investigated its importance for plasmid transmission to different *E. coli* strains. Finally, we evaluated the effect of the deletion of either the major (*pilS*) or the minor (*pilV*) pilins for conjugative transfer in the mouse intestinal tract. Our results suggest that the T4P is essential to trigger contact between donor and recipient cells in unstable environments and allows the conjugative transfer of TP114 in broth and the murine gut microbiota.

## RESULTS

**TP114 carries a thin flexible pilus which is part of the type IVb family.** TP114 comprises 11 *pil* genes distributed within two clusters (Fig. 1) but is missing the *pilI*, *pilJ*, and *pilK* genes, which are all present in related IncI1 conjugative plasmid R64 (34). To determine the main features of TP114 T4P, we started by analyzing the predicted protein sequence of PilS, annotated as the major pilin in TP114. The *pilS* open reading frame (ORF) encodes a preprotein containing a conserved PilS superfamily domain (pfam08805). A phylogenetic analysis of the major prepilin protein sequences of TP114

and other selected T4P found in various bacterial species separates them into three distinct groups that are consistent with the different T4P families, i.e., classes IVa, IVb, and IVc (Fig. S1). As expected, the PilS preprotein carried by TP114 clustered within the same class as the R64 PilS protein, corresponding to class IVb. In addition, according to the sequence alignment, TP114 PilS seems to share similar characteristics of other IVb members (3), such as a long predicted signal peptide (26 residues) and a relatively large protein (185 residues), with glutamate at position 5 of the mature pilin (Fig. S2).

To confirm the existence of TP114 T4P, we then sought to visualize it by microscopy. However, T4P are very thin (ranging between 5 to 9 nm in diameter) (3, 17, 31) compared to thick pili (ranging between 8 to 12 nm in diameter) (41) and cannot be readily visualized by conventional light microscopy (1). Consequently, we took advantage of a cysteine knock-in strategy to examine the T4P of TP114. This technique is based on the introduction of a cysteine in the major pilin subunit for subsequent labeling with a thiol-reactive fluorescent maleimide dye (32). The predicted PilS topology of TP114 (Fig. S3A) is highly similar to that of PilS from R64, and its primary sequence already contains two cysteines that presumably form a disulfide bond for stabilization (42), which are therefore not available for labeling. Based on their nucleophilic properties and their relative surface accessibility in the classical "lollipop" PilS fold (14), 10 serine or threonine residues were selected for cysteine replacement (Fig. 2A and Fig. S3A) (43). A series of single amino acid replacements were performed by recombineering with a PCR product containing the appropriate point mutation in *pilS* along with an antibiotic resistance marker to select for transformants. The cysteine knock-in mutations were introduced in both TP114 and the previously described eB-TP114 derivative (44). The introduction of cysteine in *Caulobacter crescentus* major pilin (PilS) did not apparently affect the structure of the pilus fiber or the cell envelope (45). Therefore, to evaluate the effect of the cysteine knock-in mutations in PilS activity in TP114, we performed mating experiments in broth or using a solid support. Although some cysteine knock-in mutations reduced TP114 conjugative transfer rates in broth, none completely abolished it (Fig. S3B). All individual cysteine knock-in mutants were thus used to examine the morphology of TP114 pili in *E. coli* by fluorescence microscopy using maleimide-conjugated molecules. To promote T4P expression before staining, a conjugation experiment was performed in broth with a 1:1 ratio of donor and recipient bacteria. Recipient cells were also transformed with pNeonGreen to facilitate the discrimination of the two populations. As already reported, fluorescence signal encompassing the cell body can also be detected for all bacteria since naturally occurring surface proteins can be stained as well. T4P were difficult to observe since surface-exposed filamentous fibers are often fragile and shearing is likely to occur during wash steps required for labeling (32). The orientation of the T4P in the focal planes further reduced the probability of detection by microscopy, thus preventing precise quantification. Nonetheless, structures protruding from the cells that corresponded to a thin pilus could be visualized for a few cysteine knock-in mutants, for instance, the eB-TP114 S56C mutant (Fig. 2B). No such structure was detected in *E. coli* harboring wild-type TP114 or TP114Δ*pilS*. While in most instances the observed T4P were attached to the sides of the rod-shaped cells, they were also occasionally present at the poles without any clear preference.

**Characterization of the TP114 shufflon.** Similar to other I-complex plasmids such as R64 and R721, conjugative plasmid TP114 contains a putative shufflon spanning from the distal section of the *pilV* gene to the *rci* gene encoding the shufflase (22). The constant N-terminal segment of TP114 PilV consists of 345 amino acids with a variable C-terminus that can vary from 69 to 113 residues (Fig. 3). The TP114 shufflon consists of four invertible DNA segments (instead of three as previously published in GenBank accession number MF521836.1) flanked by *sfx* recombination sites composed of a perfectly conserved 7-bp core site and 12-bp right arm, except for a 3-bp variable site located at position 7 to 9 of the right arm (Fig. S4A). The variable 3-bp GTG (e) and ATC (f) were already observed in R721, whereas TCG (g) is an additional hitherto-unseen IncI2 repeat. Alignment of the *sfx* sites also generated a consensus sequence in

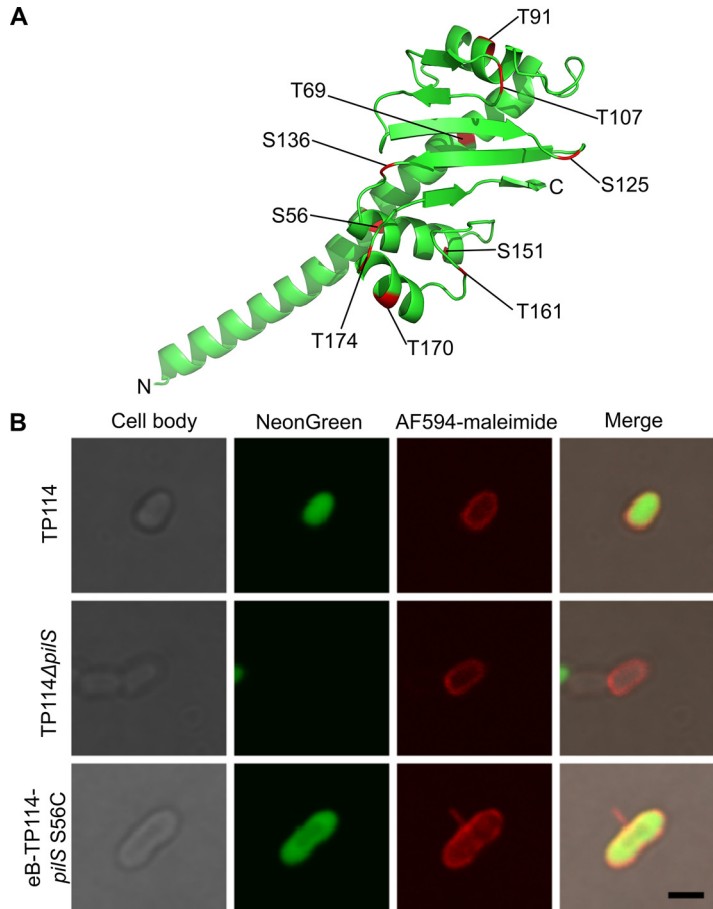

**FIG 2** The cysteine knock-in strategy allows visualization of the T4P of TP114. (A) The predicted tertiary structure of TP114 major pilin PilS. Each residue (serine [S] or threonine [T]) selected for a single cysteine knock-in is identified in red. (B) Representative images of AF594-maleimide-stained *E. coli* BW25113Δ*fliA* harboring TP114 wild type, TP114Δ*pilS*, or eB-TP114*pilS*-S56C observed by phase-contrast or widefield fluorescence microscopy. The brightness of each channel was adjusted equally between conditions. To promote the expression of the T4P, TP114-containing cells were mixed with NeonGreen expressing recipient cells before cell staining and visualization. Recipient cells that received the conjugative plasmid could then become new donors, as exemplified in the eB-TP114*pilS*-S56C mutant. Scale bar: 2 μm.

agreement with the motif found in R64 and R721 (23, 37). Each invertible DNA segment contains two convergent ORFs that can be reorganized by the shufflase to provide eight different C-terminus variants of the minor pilin PilV. By comparing the shufflon sequence with the closely related R64 and R721 plasmids, we identified three DNA segments, totaling six different ORFs (C, C', B', D', A, and A'), shared among the three plasmids, as well as another ORF corresponding to the PilVB variant of R64 (Fig. 3). Interestingly, comparative sequence analysis also revealed that TP114 possesses an additional variant not shared by the other two plasmids, referred to as D, which has, to our knowledge, not yet been reported in a publication.

Previous quantitative analyses of shufflon regions in other plasmids revealed different shufflon arrangements in clonal bacterial populations (23, 37, 46). Given the four invertible DNA segments present in TP114, a total of 384 shufflon conformations are theoretically possible (46). To investigate how the TP114 shufflon segments are dynamically reorganized or if some variants are favored over others, we first immobilized the shufflon by deleting the *rci* gene encoding the shufflase (Fig. S4B and Fig. S5A). The abundance of each *pilV* gene variant was then determined in *E. coli* cell populations carrying TP114, TP114Δ*rci*, or TP114Δ*rci* complemented in *trans* with an arabinose-inducible copy of the *rci* gene (pRCI). Illumina sequencing libraries were prepared

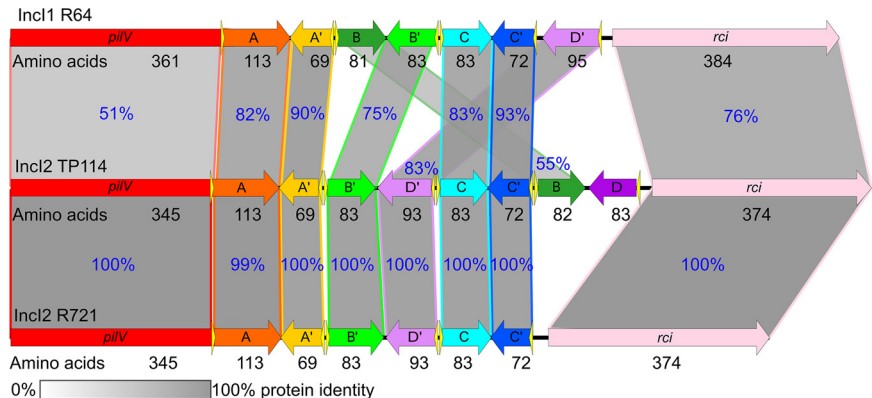

**FIG 3** Comparison of R64, TP114, and R721 shufflons. The shufflons of R64 (AP005147.1), TP114 (MF521836), and R721 (NC_002525.1) are drawn to scale. Schematic illustration showing the percent protein identity between predicted ORFs of the three different shufflons. Only one possible conformation for each shufflon is shown. Vertical blocks between sequences indicate regions of shared similarity shaded according to the percentage of identity, which is also indicated in blue for each comparison. Protein length in amino acids is indicated below each compared ORF.

with specific primers allowing amplification of the junction between the *pilV* 3′ end and the adjacent shufflon segment, forming a unique 39-bp signature for each variant (Fig. 4A to C). The abundance of each possible variant was then calculated from the raw FASTQ files. As reported previously (36), in cells harboring TP114, all possible *pilV* variants were detected, yet at different abundance levels (Fig. 4A). In contrast, only the *pilVD* variant was detected in the *E. coli* clone containing TP114Δ*rci* (Fig. 4B), suggesting that the shufflon was locked in this conformation after the loss of the shufflase. When the shufflase was reintroduced in *trans*, the shufflon became active again and the abundance of each variant slightly differed from the pattern observed in the strain containing TP114 wild type (Fig. 4C). These results confirm that the predicted shufflon is active in TP114 and that the shufflase is responsible for the rearrangement of the 3′ end of the minor pilin carrying gene *pilV*.

Transfer of TP114Δ*rci* plasmid complemented in *trans* by pRCI offers an interesting approach to analyze which *pilV* variant is more adapted for the T4P to promote conjugation in broth. In this context, the shufflon is active only in the donor strain, and *pilV* genes encoding specific PilV variants that adequately recognize recipient cells should be enriched upon sequencing of the transconjugant population. We thus performed a series of transfer assays in broth using *E. coli* MG1655Rf^R/TP114Δ*rci* + pRCI as the donor strain and three different recipient *E. coli* strains, namely, MG1655Nx^R, Nissle1917, and J96. Interestingly, specific variants of *pilV* were found enriched upon transfer in comparison to the donor strain population (Fig. 4D). For example, only *pilVB′* was found to be clearly enriched after conjugation with *E. coli* J96. Furthermore, the resulting conjugation profiles were considerably different between recipient strains, suggesting that only certain *pilV* variants can adequately recognize the surface molecules exposed by the tested strains and allow efficient transfer of TP114.

**TP114 PilV variants specificity.** To further study the specificity of each PilV variant, eight TP114 derivatives were constructed, in which the whole shufflon was replaced by a single *pilV* gene variant expressed from its endogenous locus (Fig. S5). Each derivative was then subjected to pairwise mating experiments in broth with the same three *E. coli* recipient strains tested previously (MG1655Nx^R, Nissle1917, and J96). As expected, the observed conjugation frequencies (Fig. 4E) were well correlated with the enrichments observed for conjugation of pooled variants (Fig. 4D), with the exception for PilVC′ and PilVA′ that showed only limited enrichment in MG1655Nx^R and J96, respectively. High-throughput transfer assays in broth were next performed to increase the diversity of biological structures presented to the PilV variants library and further investigate the specificity of each variant toward different recipient cells. A collection of various *E. coli* serotypes

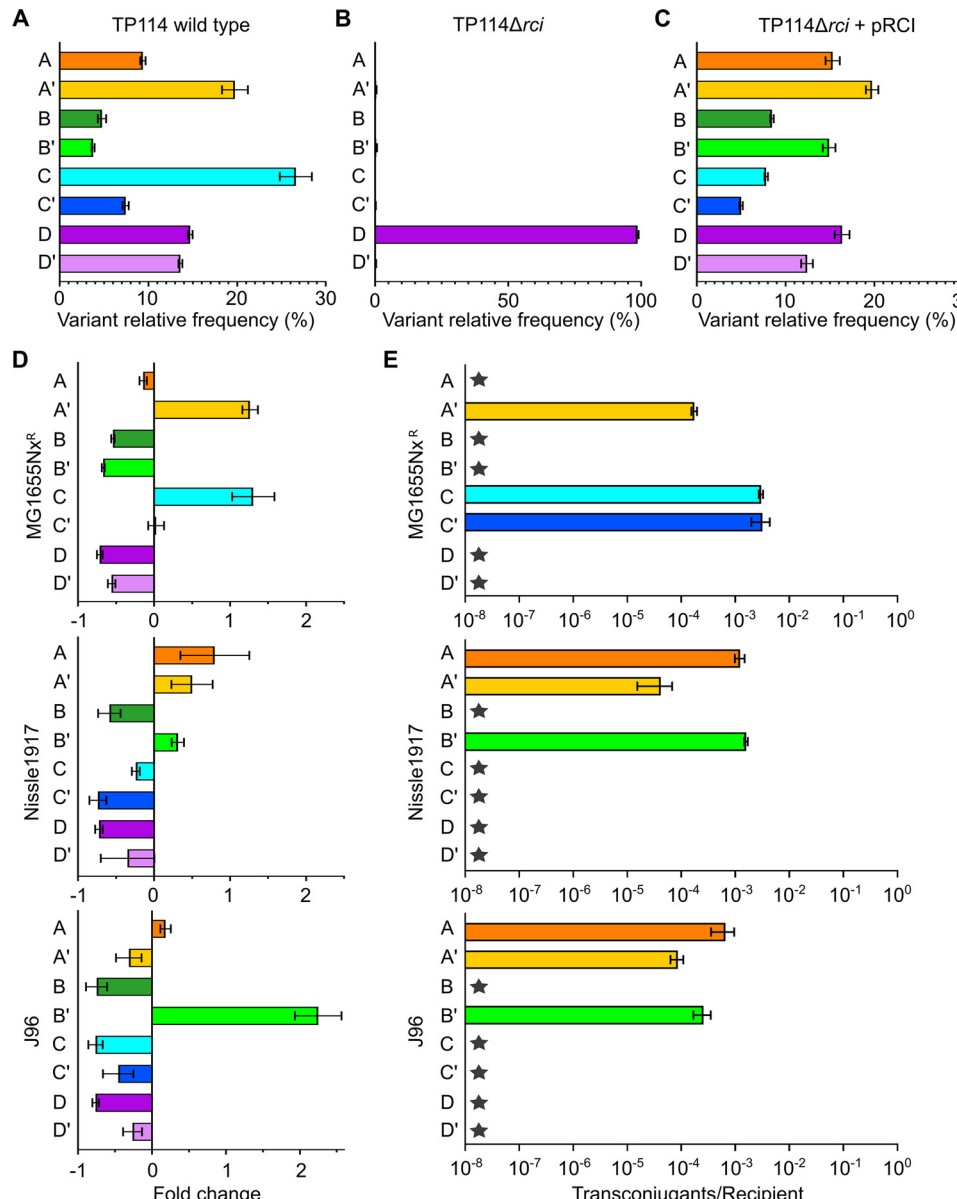

**FIG 4** The 3' end of the *pilV* gene is subject to dynamic rearrangements by the TP114 shufflase. (A to C) Relative *pilV* variant frequencies, determined by Illumina sequencing, for *E. coli* MG1655Rf^R harboring TP114 (A), TP114Δ*rci* (B), or TP114Δ*rci* complemented in *trans* with pRCI (C). Bars and error bars represent the mean and standard deviation calculated from biological triplicates, respectively. (D) Enrichment of *pilV* variants, determined by Illumina sequencing, after conjugation of TP114Δ*rci* from MG1655Rf^R to *E. coli* MG1655Nx^R, Nissle1917, or J96. (E) Transfer frequencies of each variant individually fixed at the 3' end of the *pilV* gene (see Fig. S5B) toward three different *E. coli* recipient strains. Stars mean transfer rates below the limit of detection of the experiment ($1 \times 10^{-8}$).

was tested, including 13 recently isolated swine isolates, for a total of 27 different strains (see Table S1). As shown in Fig. 5, each PilV variant displayed a distinct conjugation pattern, with occasional overlaps in their specificities (e.g., C' versus B'). These results also confirmed that all TP114 PilV variants are functional in broth, each recognizing between 8 and 24 of 27 tested recipient strains and contributing to the overall pattern observed for TP114 wild type (Fig. 5). To confirm that the variable domain of PilV is the actual region responsible for the recognition of cell surface molecules, we also replaced this segment with a FLAG tag (TP114Δ*pilV*::FLAG-*cat*, see Fig. S5D) and subjected this construct to the same high-throughput screening. As expected, while the conjugation profile of this mutant was virtually unaffected on solid medium, its capacity to conjugate in broth was

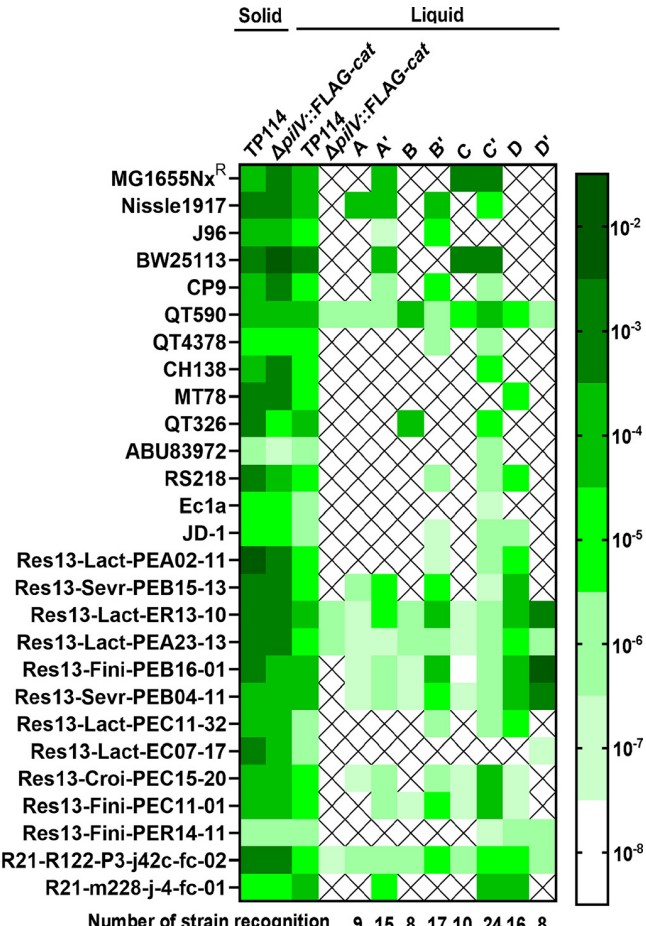

**FIG 5** Conjugation frequencies of TP114 *pilV* variants toward various *E. coli* recipient cells. Heat map illustrating the transfer rates of TP114 wild type and the eight fixed *pilV* variants for different strains and serotypes of *E. coli*. Transfer rates of the TP114Δ*pilV*::FLAG-*cat* mutant in which the variable region of *pilV* was replaced by a FLAG tag are also shown. All conjugations were performed in biological triplicate with KN01Δ*dapA* (Nissle1917Δ*dapA*, Sp$^R$, Sm$^R$) as the donor strain. Cross marks indicate conjugation frequencies below the detection limit of the experiment ($1 \times 10^{-8}$). The number of strains recognized by a given variant is indicated below the heat map.

severely altered, showing transfer rates just above the limit of detection for only four strains (Fig. 5). These residual conjugation rates could be explained by the presence in the recipient strains of plasmids or genes that contribute to mating pair stabilization.

**Both major and minor pilins are required for conjugation in the mouse microbiota.** To confirm that the major (PilS) and minor (PilV) pilins are critical for mating pair stabilization and transfer of T4P *in vitro*, both genes were deleted individually (Fig. S6) and conjugation experiments were performed under various conditions. As expected, the absence of either pilin did not significantly impede the capacity of the plasmid to transfer between *E. coli* Nissle1917 cells on a solid support (Fig. 6A). However, deletion of *pilS* or *pilV* abolished the transfer of TP114 in static or shaking broth conditions, confirming their essential role in the T4P. Replacement of the variable 3′ end of *pilV* by a FLAG tag resulted in the same phenotype, providing additional evidence of the critical role of the PilV C-terminus in the recognition of the recipient cell and stabilization of the mating pair. Importantly, transfer rates can be partially restored upon complementation with the corresponding pilin provided in *trans* on pBAD30-derived plasmids (pPilS or pPilVA) (Fig. 6A).

To evaluate the impact of these deletions in a more natural environment, we next performed *in vivo* conjugation assays directly in the mouse gut microbiota (40) and measured transfer rates in feces and cecum. While TP114 wild type transferred at high

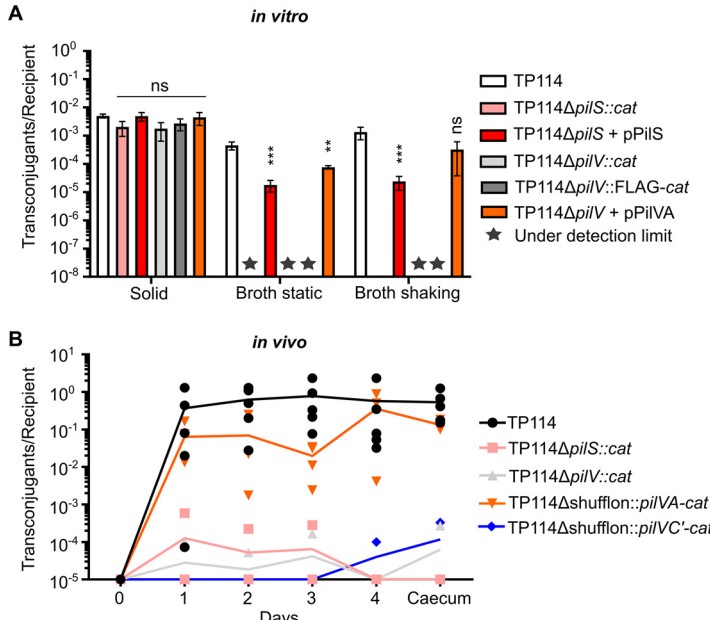

**FIG 6** Transfer rates of TP114 major or minor pilin deletion mutants *in vitro* and *in vivo*. (A) *In vitro* conjugation rates of TP114 wild type, or deletion mutants, complemented or not, are shown in different conditions. Stars indicate transfer rates under the limit of detection of the experiment. Bars and error bars represent the mean and standard deviation calculated from biological triplicate, respectively. One-way ANOVA results are shown. ns, $P \geq 0.05$; *, $P < 0.05$; **, $P \leq 0.01$; ***, $P \leq 0.001$. (B) Conjugation frequencies of TP114 wild type as well as *pilS* and *pilV* deletion mutants in the mouse gut microbiota evaluated from the feces collected at different time points. Transfer rates of TP114 lacking the entire shufflon but expressing the *pilVA* or the *pilVC'* variants are also shown. Each symbol represents the value of one mouse and the lines represent the mean for each different condition across the experiment time. The limit of detection of the experiment corresponds to $1 \times 10^{-5}$ transconjugants per recipient.

frequencies after the successive introduction of the recipient and donor *E. coli* Nissle1917 strains by gavage, the *pilS* and *pilV* deletion mutants were highly affected, showing transfer rates just above the detection limit (Fig. 6B). Moreover, the complementation of TP114Δ*pilV* with a *pilVA* variant recognizing *E. coli* Nissle1917 restored the phenotype close to the levels observed with TP114 wild type (Fig. 6B). In contrast, the *pilVC'* variant showed low levels of conjugation in the cecum and feces, as expected given its low ability to support conjugation in broth. Taken together, these results demonstrate that the presence of both PilS and a competent PilV variant is essential to the formation of a functional T4P allowing transfer in unstable environments such as the mouse gut microbiota.

## DISCUSSION

In this work, we dissected the essential role of the T4P carried by the IncI2 conjugative plasmid TP114 for mating pair stabilization in broth and unstable environments such as the gastrointestinal tract. We first sought to confirm the presence of the T4P at the surface of TP114 harboring bacteria using a cysteine knock-in strategy (Fig. 2 and Fig. S3A). However, the observation of pili was difficult, which can be explained by different reasons. For example, only a subpopulation of bacteria could be expressing a pilus as previously reported for the type IVb pilus carried by the pathogenicity island PAPI-1 of *P. aeruginosa* (47). Also, even under optimal conditions, only a fraction of the donor bacteria population display conjugative transfer competence in *E. coli* (48). It was reported that no more than 0.1 to 1% of mating pairs are engaged in transferring DNA at any given moment (49). In the case of F-like plasmids, approximately 1% of donor cells activate their *tra* genes and become transfer competent cells, except for derepressed plasmids (50). These observations underline that conjugation is a highly energy-consuming process that cells cannot continuously support with the synthesis

of complex structures such as the T4P. Deciphering the regulatory mechanisms controlling the expression and activity of the TP114 T4P will thus represent an interesting topic for future research.

Like other members of the I-complex, TP114 contains a multiple DNA inversion system called shufflon. Originally described in IncI1 R64 and later in IncI2 R721 conjugative plasmids, the shufflon is controlled by a tyrosine recombinase (shufflase) that swaps the 3′ end of the minor pilin-coding gene *pilV* while maintaining the highly conserved N-terminus region (22). TP114 carries eight variable 3′-end segments of *pilV* adhesins (Fig. 3), which is to our knowledge the first characterized shufflon containing more than seven different *pilV* partial ORFs. Interestingly, our experiments performed with a donor strain harboring TP114Δ*rci* complemented in *trans* with an inducible shufflase expressed from plasmid pRCI revealed the efficiency and rapidity of the shufflon to select the appropriate variants and allow conjugative transfer with different recipient strains (Fig. 4). In this context, the donor and recipient strains (*E. coli* MG1655Nx$^R$, Nissle1917, or J96) were put in contact for a maximum of 20 min before plating the conjugation mix on a selective medium to obtain transconjugants. Since the shufflase was present only in the donor strain, the shufflon structure recovered in the transconjugants was fixed in the conformation used at the moment of transfer. The results showed that specific *pilV* variants were enriched in the transconjugant populations and differed for each recipient strain. Interestingly, the enriched variants obtained in this experiment (Fig. 4D) were found to be the same that allowed conjugative transfer in broth (Fig. 4E). We also noticed that the pRCI arabinose-inducible shufflase expression plasmid allowed the reorganization of the shufflon under repressing conditions and that overexpression of the shufflase resulted in a 3-orders-of-magnitude decrease in viability of the donor bacteria. These observations suggest that very low concentrations of the shufflase are required to constantly reorganize the shufflon, which could facilitate the dissemination of TP114 with a diversity of recipient strains.

We also performed systematic conjugation assays with each of the eight immobilized *pilV* variants (Fig. S5B) in a shufflase-deficient strain to investigate their functionality and impact on conjugative transfer in broth. A panel of 27 *E. coli* strains was screened (Fig. 5), revealing that all PilV variants of TP114 are biologically active in broth mating. The PilV variants recognized different sets of strains, suggesting that the molecular structures they recognized are distinct. These interactions clearly involve the C-terminal binding domain of PilV since the TP114Δ*pilV*::FLAG-*cat* mutant could no longer be transferred in broth, except at low rates toward recipient strains that appear to be capable of mating pair stabilization. In plasmid R64, the adhesins formed by the different *pilV* products were shown to bind specifically to distinct lipopolysaccharide structures at the surface of recipient cells, thus determining the recipient specificity in broth conditions (6, 22). For example, it has been reported that the ligands of PilVA, PilVB′, PilVC, and PilVC′ adhesins are *N*-acetylglucosamine-*β*-(1-3)-glucose, *N*-acetylglucosamine-*α*-(1-2)-glucose, *N*-acetylglucosamine-*β*-(1-7)-heptose, and glucose-*α*-(1-2)-glucose or glucose-*α*-(1-2)-galactose, respectively (6). Given the high protein sequence similarity between R64 and TP114 PilV C-terminus segments (Fig. 3), it would be interesting to determine if these variants recognize the same structures. Additional work will be needed to identify the exact ligands recognized by each of the eight TP114 PilV variants. The specificity of the PilVD variant will be particularly interesting to study, as it is found in TP114 but not in other described shufflons. Nonetheless, our results demonstrate that each of the *pilV* variants present in the TP114 shufflon allows the recognition of different surface receptors, which together contribute to the establishment of an expanded transfer host range.

Unravelling how T4P sticks to diverse surfaces and if adherence occurs only at the tip of the fibers or along the entire structure (14) would particularly help to better understand the interactions involved between the T4P and the surface of recipient cells. While some minor pilins (CofB) have been located at the pilus tip (25–29), this question remains unanswered for the majority of T4P. The number of different

adhesins and their position in a pilus might also affect the recognition of certain receptors. Structural studies could have a major role in addressing these aspects.

Essential genes involved in the conjugative transfer of TP114 in different conditions have been identified previously by high-density transposon mutagenesis (40). Interestingly, 9 of 11 *pil* genes involved in T4P biogenesis were shown to be indispensable for productive mating pair formation and stabilization in the gut microbiota (Fig. 1). To further validate that the T4P is essential for mating pair stabilization in broth mating conditions, we inactivated key genes implicated in the pilus structure (Fig. S6). The ability of TP114Δ*pilS* and TP114Δ*pilV* mutants to transfer was assessed using conjugation assays performed in broth or on solid support (Fig. 6A). We corroborated the essential role of the major and minor pilins since the inactivation of either of those genes impaired the function of the T4P only for matings in broth. A standardized mouse model (40) was used to further demonstrate the importance of the T4P harboring an appropriate PilV adhesin in enabling conjugative transfer in the gut microbiota (Fig. 6B).

For many years now, antibiotic resistance has been recognized as a major problem for the public health system (51). T4P are essential virulence factors in numerous human pathogens (17, 27, 29, 52) and contribute to antimicrobial resistance gene dissemination. These extracellular appendages, therefore, represent potential targets to develop novel vaccines or antiadhesive therapies to fight bacterial infections and reduce the dissemination of antibiotic resistance genes.

## MATERIALS AND METHODS

**Bacterial strains and growth conditions.** Bacterial strains and plasmids used in this study are listed in Table S1. Routinely, bacterial strains were grown at 37°C in Miller's lysogeny broth (LB) in an orbital shaker/incubator set at 225 rpm or on LB agar in a static incubator. All strains were preserved at −80°C in LB broth containing 25% (vol/vol) glycerol. Cells with thermosensitive plasmids (pSIM6, pE-FLP) were grown at 30°C. When appropriate, antibiotics were used at the following concentrations: 100 $\mu$g/mL ampicillin (Ap), 34 $\mu$g/mL chloramphenicol (Cm), 50 $\mu$g/mL kanamycin (Km), 4 $\mu$g/mL nalidixic acid (Nx), 66.7 $\mu$g/mL rifampicin (Rf), 100 $\mu$g/mL spectinomycin (Sp), 50 $\mu$g/mL streptomycin (Sm), and 15 $\mu$g/mL tetracycline (Tc). Diaminopimelic acid (DAP) auxotrophy was complemented by adding DAP at a final concentration of 57 $\mu$g/mL in the medium. To induce expression from $P_{BAD}$ promoter, LB medium was supplemented with 0.1 to 1.0% L-arabinose.

**Molecular biology.** Plasmid DNA was prepared using the EZ-10 spin column plasmid DNA miniprep kit (Bio Basic), whereas genomic DNA (gDNA) for Illumina sequencing libraries was prepared using the Quick-DNA magbead plus kit (Zymo Research) according to manufacturer's instructions. Restriction enzymes were purchased from New England Biolabs. PCR amplifications were performed using TransStart FastPfu Fly DNA polymerase (Civic Bioscience) or TaqB (Enzymatics) for DNA amplifications and PCR screening, respectively. PCR products were purified using MagicPure DNA size selection beads (Civic Bioscience) or Monarch PCR and DNA cleanup kit (New England Biolabs) following the manufacturer's recommendations before assembly or recombineering, respectively. Sequences of interest were confirmed by Sanger sequencing at the Plateforme de séquençage et de génotypage du Centre de Recherche du CHUL (Université Laval, QC, Canada).

**Plasmid and strain construction.** A detailed list of oligonucleotides used in this study and their usage is found in Table S2. Plasmids were assembled from purified PCR products or digested DNA fragments using the 2× NEBuilder Hifi DNA assembly master mix (New England Biolabs) according to the manufacturer's protocol. Following assembly, constructs were treated 30 min at 37°C with DpnI to eliminate residual DNA templates before transformation into chemically competent *E. coli* EC100D*pir*+ as described by Green and Rogers (53). Plasmids were then introduced into bacterial strains of interest by electroporation as described in Nováková et al. (54) using a Bio-Rad GenePulser Xcell apparatus set at 25 $\mu$F, 200 $\Omega$, and 1.8 kV with 1-mm gap electroporation cuvettes. Deletion mutants as well as TP114 cysteine knock-in mutants were generated by recombineering using pSIM6 plasmid and 40-bp homology extensions as described previously (40, 55, 56). When required, the introduced antibiotic resistance cassette was removed from the resulting construction by Flp-catalyzed excision using pE-Flp (57). All modifications were verified by PCR and Sanger sequencing.

**Fluorescence microscopy.** Flagella-deficient *E. coli* BW25113Δ*fliA* from the Keio collection (58) was used for T4P labeling to avoid potential interference from the flagellum. A model of the tertiary structure of the major pilin PilS was predicted by I-TASSER (59, 60) and visualized by PyMOL (The PyMOL Molecular Graphics System, Version 1.2r3pre, Schrödinger, LLC). In total, 10 serine or threonine residues, predicted to be on the surface of the protein PilS according to the NetSurfP software v2.0 (43), were selected for single mutation into cysteine. Visualization of the T4P was done by maleimide-conjugated fluorophores as described by Ellison et al. (32). Before the observations, *E. coli* BW25113Δ*fliA* cells harboring either TP114 or TP114 derivatives were subjected to conjugation in broth with *E. coli* BW25113Δ*fliA* recipient cells carrying pNeonGreen. Conjugations in broth were performed as described

for *in vitro* conjugation experiments, except that cells were harvested after 5 min of mating for eB-TP114 (44) plasmid or after 70 min for TP114 wild type. Cells were then centrifuged for 1 min at $400 \times g$ and resuspended in sterile phosphate-buffered saline (PBS) $1\times$. Fluorophore-conjugated maleimide dye (Alexa Fluor 594 $C_5$-maleimide [A10256, Thermo Fisher Scientific]) was added to the cells at a final concentration of 25 $\mu$g/mL. Labeling was performed at 37°C for an interval of time ranging between 30 min and 4 h. Labeled cells were then washed twice with PBS $1\times$. Cells were deposited on an agarose pad and then covered with a glass coverslip. Cell bodies were photographed using phase-contrast microscopy, whereas labeled pili were observed using widefield fluorescence microscopy. All images were acquired using an Axio Observer Z1 inverted microscope (Zeiss) equipped with a $100\times$ Plan Apo oil objective (numerical aperture = 1.4), a 470/40 excitation filter, a 525/50 emission filter, and an AxioCam 506 mono camera (Zeiss). Images were captured using the Fluoview 2.1 imaging software and analyzed using Fiji (61).

**In vitro conjugation assays.** All *in vitro* conjugation assays used *E. coli* Nissle1917 (KN01Δ*dapA*) and *E. coli* Nissle1917 (KN03) as the donor and recipient strains, respectively, except for fluorescence microscopy and shufflon mating experiments (see corresponding Materials and Methods sections). Conjugation assays were performed as described in Neil et al. (40). Conjugation frequencies were calculated according to the number of transconjugants per recipient CFU. To determine the conjugation frequencies with individual *pilV* variants toward various bacterial strains, high-throughput conjugations in broth were performed using *E. coli* KN01Δ*dapA* as the donor. All tested *E. coli* recipient strains are listed in Table S1. In that context, strains were grown for 18 h without any antibiotic pressure. Volumes of 30 $\mu$L donors and 20 $\mu$L recipients were then directly mixed in a 96-well microplate containing 150 $\mu$L LB broth supplemented with DAP before incubating for 2 h at 37°C. After the incubation period, mating cells were serially diluted in PBS $1\times$ and spotted in triplicates on LB agar selective plates containing the appropriate antibiotic(s) to discriminate donors (Sp, DAP, Kn, Cm), recipients (no antibiotic), or transconjugants (Kn, Cm). For experiments with strains containing an arabinose-inducible plasmid (pPilS, pPilVA), 1.0% L-arabinose was added to the cultures after 16 h of growth and was maintained during the conjugation assays. All experiments were performed in biological triplicate using three independently grown cultures.

**Shufflon mating experiments.** To evaluate the abundance of PilV variants in the context of the active TP114 shufflon, conjugation experiments were carried out using *E. coli* MG1655Rf^R harboring either TP114 or TP114Δ*rci* complemented or not with pRCI as the donor strain. Donor strains were cultured 16 h in LB broth containing 1% glucose and appropriate antibiotics. Then, 0.1% L-arabinose was added to the cultures for $P_{BAD}$ induction and cells were incubated at 37°C for 30 min. As high induction of the shufflase appears to have a negative impact on strain fitness, cells were then centrifuged 5 min at $2,100 \times g$ before suspension in 5 mL of LB broth with 1% glucose and appropriate antibiotics. Cells were allowed to recuperate for 2 h at 37°C before conjugation assays. Different *E. coli* recipient strains were selected based on their antibiotic resistance profile to allow proper selection of donors (Rf, Kn, and Ap when pRCI was used), recipients (Nx for MG1655Nx^R and J96 or Sm or Tc for KN03), and transconjugants (Kn and antibiotic[s] selecting the appropriate recipient strain) after mating. Donor and recipient strains were mixed at a 1:1 ratio according to the measured optical density at 600 nm (OD$_{600}$) and allowed to conjugate in broth for between 10 to 20 min to avoid plasmid retrotransfer. All experiments were performed in biological triplicate using three independently grown cultures. Samples of 40 $\mu$L of the donor strains were collected just before mating experiments and kept at $-20$°C until gDNA extraction. Transconjugants were recovered by scraping appropriate selective plates and used for gDNA extraction.

**In vivo conjugation assays.** All mice-related protocols were performed as described by Neil et al. (40) and strictly followed the Université de Sherbrooke Animal Care Committee Guidelines. C57BL/6 female mice of 16 to 20 g (Charles River) were used for this study. Mice were housed in individually ventilated cages with no more than five individuals per cage. Animals were given at least 3 days of rest upon arrival and were provided with water and standard chow (Charles River) *ad libitum*. A concentration of 1 g/L of streptomycin was added to drinking water 2 days before gavage to deplete endogenous populations of enterobacteria in the digestive tract and to facilitate the establishment of streptomycin-resistant *E. coli* Nissle1917 strains. Mice were orally challenged with the recipient strain *E. coli* Nissle1917 Sm^R Tc^R (KN03) 2 h before the introduction of the donor strain *E. coli* Nissle1917Δ*dapA* Sp^R Sm^R (KN01Δ*dapA*) containing the appropriate plasmids. Conjugation was monitored in feces samples daily as described by Neil et al. (40). Mice were sacrificed on the fourth day and the cecum was extracted to measure conjugation levels in the murine gut. Feces were homogenized and CFU were counted on MacConkey selective plates to discriminate donor (Sp, Sm, Kn, and DAP), recipient (Sm, Tc), and exconjugants (Sm, Tc, and Kn). All *in vivo* experiments were performed using 3 to 5 mice.

**Phylogenetic analysis.** The NCBI database was used to retrieve the amino acid sequence of major pilins listed in Fig. S2. Briefly, selected pilin protein sequences from different bacteria were aligned using the MUSCLE v3.8.31 algorithm (62) in the SeaView 5.0.4 phylogenetic analysis software package (63). The TP114 PilS signal peptide cleavage site was determined by visual inspection of the alignment and according to the data available for the other pilins. The maximum-likelihood phylogenetic tree was generated from the alignment file according to the PhyML method with an LG 4-rate class model (64). Branch-support values were calculated by bootstrap analysis using 1,000 replicates.

**Shufflon sequencing and analysis.** Reanalysis of the TP114 sequence (MF521836.1) and new sequencing data obtained from the TP114Δ*rci* mutant revealed a new segment in the shufflon region (named D). Therefore, a new file including the whole shufflon sequence has been resubmitted to GenBank (MF521836.2) and was used in this study. Schematic representations of R64 (AP005147.1), TP114, and R721 (NC_002525.1) shufflons were generated by EasyFig (65), and the percent protein

identity was determined by BLASTp optimize for highly similar sequences (megablast) using default search parameters. Sanger chromatogram of the 3′ end constant region of *pilV* was generated previously (40) and was analyzed using CLC Main Workbench. Multiple alignments and corresponding consensus sequence of the eight repeat regions (*sfx* sites) of the TP114 shufflon were generated by the WebLogo server (https://weblogo.berkeley.edu/logo.cgi) (see Fig. S4A).

**Illumina sequencing library preparation.** DNA libraries were prepared from extracted gDNA using the NEBNext Ultra II FS DNA library prep kit for Illumina (New England Biolabs) following the manufacturer's instructions with some modifications. Adaptors ligation was performed using preannealed oSh-F and oSh-R primers (Table S2) at a working concentration of 2 $\mu$M. Size selection and cleanup of adaptor-ligated DNA were done using AMPure XP beads (Beckman Coulter). Libraries were amplified by quantitative PCR using Veraseq 2.0 high-fidelity DNA polymerase (Enzymatics) and Illumina i5 and i7 index primers with the following protocol: (i) 30 sec at 98℃, (ii) 35 cycles of 15 sec at 98℃, 30 sec at 60℃, and 18 sec at 72℃, and (iii) 2 min at 72℃. Reactions were stopped before reaching the amplification plateau. Library quality was assessed on a 5200 fragment analyzer instrument (Agilent) and DNA concentration was measured using the Quant-iT PicoGreen dsDNA assay kit (Thermo Fisher). A total of 15 different libraries were prepared, in 3 biological replicates. Illumina sequencing was performed by the McGill Genome Center using an Illumina NovaSeq instrument.

**Bioinformatic analysis.** Unix scripts were used to count the occurrence of each 39-bp unique footprint corresponding to the different PilV variants directly from the raw FASTQ files. The relative abundance of a given variant was computed as the number of reads containing the corresponding footprint divided by the total number of reads positive for any variant footprint.

**Statistical analyses.** Statistical significance tests were performed on the logarithmic values of data using one-way analysis of variance (ANOVA). Differences between control and test data groups were considered significant when the calculated $P$ value was below 0.05. Statistical significance was noted directly on the figures as follows: ns, $P \geq 0.05$; *, $P < 0.05$; **, $P \leq 0.01$; ***, $P \leq 0.001$.

**Data availability.** The sequence of TP114 used in this study was deposited on GenBank (MF521836.2).

## SUPPLEMENTAL MATERIAL

Supplemental material is available online only.
**SUPPLEMENTAL FILE 1**, PDF file, 3.1 MB.

## ACKNOWLEDGMENTS

This work was supported by the Canadian Institutes of Health Research (CIHR no. 159817). S.R. holds a Chercheur boursier senior fellowship from the Fonds de recherche du Québec - Santé (FRQS). N.A. is supported by a doctoral scholarship from the Université de Sherbrooke. K.N. is the recipient of a Fonds de Recherche du Québec - Nature et Technologies (FRQNT) doctoral fellowship and is supported by the Natural Science and Engineering Research Council of Canada (NSERC). Funding for open access charge: CIHR #159817.

Some aspects of the work presented in this study are part of patent application WO2020010452A1. All authors of the present manuscript except for F.G. are also coauthors of this provisional patent application. S.R. and K.N. have a financial interest in TATUM bioscience.

We thank Calcul Québec (www.calculquebec.ca) and Compute Canada (www.computecanada.ca) for access to bioinformatic resources and support. We thank the McGill Genome Center for assistance with Illumina NovaSeq sequencing. We thank Daniel Garneau, David Kysela, and Yves Brun for technical support and insights with microscopy experiments and image analysis. We also thank Nicolas Allaire Tanguay for the design and construction of plasmid pRCI. We thank Dominick Matteau for his insightful comments on the manuscript.

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
