## [Reviewer comments · Microbiology Spectrum]

Microbiology Spectrum

The type IV pilus of plasmid TP114 displays adhesins conferring conjugation specificity and is important for DNA transfer in the mouse gut microbiota.

Nancy Allard, Kevin Neil, Frédéric Grenier, and Sébastien Rodrigue

Corresponding Author(s): Sébastien Rodrigue, Université de Sherbrooke

Review Timeline:

Submission Date:	November 17, 2021
Editorial Decision:	December 30, 2021
Revision Received:	February 9, 2022
Accepted:	February 16, 2022

Editor: Jennifer Auchtung

Reviewer(s): The reviewers have opted to remain anonymous.

Transaction Report:

DOI: <https://doi.org/10.1128/spectrum.02303-21>

December 30, 2021

Prof. Sébastien Rodrigue
Université de Sherbrooke
Biologie
2500 Boulevard Université
Sherbrooke, Quebec J1K 2R1
Canada

Re: Spectrum02303-21 (The type IV pilus of plasmid TP114 displays adhesins conferring conjugation specificity and is important for DNA transfer in the mouse gut microbiota.)

Dear Prof. Sébastien Rodrigue:

Thank you for submitting your manuscript to Microbiology Spectrum. Both reviewers thought that the overall conclusions of your manuscript were well supported by your studies, but that the manuscript requires the revisions indicated below. Please address all reviewer comments below in your revisions. When submitting the revised version of your paper, please provide (1) point-by-point responses to the issues raised by the reviewers as file type "Response to Reviewers," not in your cover letter, and (2) a PDF file that indicates the changes from the original submission (by highlighting or underlining the changes) as file type "Marked Up Manuscript - For Review Only". Please use this link to submit your revised manuscript - we strongly recommend that you submit your paper within the next 60 days or reach out to me. Detailed instructions on submitting your revised paper are below.

Link Not Available

Sincerely,

Jennifer Auchtung

Journals Department
Reviewer comments:

Reviewer #1 (Comments for the Author):

This manuscript describes experiments to look at the thin pilus and shufflon in an IncI2 plasmid, including visualising the pilus by fluorescence microscopy, conjugation experiments using constructed deletion mutants and different shufflon arrangements and different recipients. The experiments appear well thought out and the conclusions are generally supported by the data, but some parts of the manuscript are hard to follow, as necessary details are missing and need to be added. The organisation/wording needs improving in places to increase accuracy and clarity.

1) TP114, Fig. 1.

The origin of TP114 is not clear. It seems to be a historic IncI2 plasmid? Brief details of this should be included and, if possible, a reference cited. It is not clear what "DSM-42426 (DSMZ) in Table S1 refers to - this needs to be explained. Also, what is the relevance of the behaviour of this plasmid? e.g. how does it relate to other Inc2 plasmid carrying important resistance genes?

In Fig. 1 genes in yellow ("selection") include "aph-III", which I think should be aph(3')-I (from a search of GenBank with the protein sequence). Presumably the kanamycin resistance conferred by this gene is what was used to select for transconjugants carrying TP114, but this doesn't seem to be clearly stated anywhere. This resistance gene needs to be made more obvious in Fig. 1 and reference to kanamycin resistance needs to be added to Table S1 and something added to the text to explain the selection.

It is not clear what other genes in yellow labelled e.g., "yadA", "077", "079", "081" have to do with "selection", e.g., "077" is listed as "unknown function" in Supplementary Data 1 of Neil et al. Please rethink the colours used for these genes.

2) Inc I numbers

The "1" of IncI1 and the "2" of IncI2 are written as subscripts throughout this manuscript, which is not conventional - see PDFs of Refs 69, 70, 72, 73, etc. This needs to be corrected, including in figures and tables, relevant reference titles and supplementary material.

3) Organisation etc.

The length of both the Introduction and Discussion should be reduced, by removing detail that is less relevant to this manuscript and reducing repetition with the Results. Over 100 references is excessive for a non-review paper and these need to be rationalised. For example, lines 42-48 include a large number of references - could a couple of reviews be used instead? - and the information in lines 49-57 is not that relevant to the remainder of the manuscript and cites a lot of references.

Lines 296-310 - it may be possible to reduce the detail needed here by referring to manufacturers' instructions and/or previous papers.

Also, some long blocks of text could be divided into paragraphs e.g. new paragraph at the end of line 469.

4) Descriptions of shufflons Lines 112-129, 397 -

Descriptions of shufflons and how they work are not clear/accurate - suggest something like "that can rearrange, changing the end..." on line 112, "rearrangement of shufflon segments, switching the 3' end of pilV" on line 120 (references are also needed here) and "switch changes the C-terminal end of PilV" on line 126, but repetition in this section also needs to be reduced. Lines 397, 408 etc - suggest something simpler, such as "different shufflon arrangements" rather than "conformational heterogeneity".

Line 420 - suggest replacing with something like "pilV genes encoding PilV variants that..."

Line 504-5 - is "controlled" the right word here and is there evidence that this is random?

Line 508 - suggest "more than seven different partial ORFs"

5) Conjugation experiments, Lines 148-155, 195, 209-224

A lot of the Methods section is taken up with descriptions of different conjugation experiments. There seems to be some repetition and it is not clear why there are all these different variations, what would be used for selection for each experiment and the relevance of DAP.

Lines 134, 136 - "host range" may suggest different species, but all recipients used here seem to be E. coli? Only E. coli are listed in Table S1, so what is this "broad range of Enterobacteriaceae"?

Line 222 - cells were washed off plates into PBS? What about the liquid cultures?

Line 234 - what were the "appropriate antibiotics" for selecting. No antibiotic resistance is listed for most strains in Table S1/Fig. 5 and these must be added to understand how these recipients were selected for.

Line 269 - how was conjugation monitored? What selection was used?

5) Homology, line 180, line 288, Fig. 3, line 939

Homology cannot be quantified, so it is incorrect to use "the percentage of homology". % identity must be used for nucleotide sequences while % identity or % similarity can be used when comparing protein sequences, depending on what has been calculated.

Line 180 - are these 40 bp regions actually identical?

6) Lines 325-416

How does this relate to what is already known for other IncI2 plasmids such as R721? There are papers on the shufflon of R721. What new information is provided here? This is not really clear.

Lines 21, 395 - searches with this segment suggest that it is present in many other plasmid sequences in GenBank - do you mean not yet described in a paper? This needs to be explained more accurately.

Lines 326-8 - it would seem more relevant to compare with another IncI2 plasmid here, rather than IncI1 R64, as the two types are not that closely related. Also line 390.

Lines 334-5 - why is this interesting? Isn't this to be expected and TP114 PilS is closer to PilS R721, the other IncI2 plasmid, than to R64.

Line 346 - how is this known?

Line 349 - this needs a reference.

Line 352 - this needs more explanation.

Lines 392, 543 - the "B" segments of R64 and TP114 are not that closely related - 55% identity according to Fig. 3, which is not that high, and line 543 says >75% and the D ORF is not related.

7) Other scientific points

Lines 77-8 - "PilS/A/E" and "PilV/E/W/X" need some explanation i.e. that these subunits have different names in different systems.

Lines 94, 326 - genes are e.g. carried, not encoded - they encode proteins. Also, lines 377, 502 - the shufflon is not "encoded" and "encoded for" is like saying "coded for".

Line 116 - the term "I complex" could be introduced here.

Lines 236, 244 - why 1.0% arabinose in one place and 0.1% in another? To only induce a low level of Rci?

Line 287 - should this be MF521836. 2? It would be good if this GenBank entry included a note explaining that the order of shufflon segments is only one possible (the dominant?) arrangement- see e.g., GenBank entries for R64, R721.

Lines 290-92 - "Sanger trace/chromatogram"? It is not clear why this was done?

Lines 405-6 - this is not clear. Illumina sequencing was performed on extractions of whole genomic DNA and then the 39 bp signatures were identified in the raw reads? This needs to be rewritten.

8) Figures

Fig. 1 - see comments above

Fig. 2A is not that relevant to the main manuscript and seems to show the same information as Fig. S3A.

Fig. 3 - the accession numbers for R64 and R721 should be added. See comments on homology above.

Fig. 5 - numbers of strains, rather than %, should really be used here, as the total is below 100 (also line 447).

Fig 6 legend - the amount of experimental detail can be reduced - this should all be in Methods or Results.

9) Minor points/English etc

Throughout - care needs to be taken when writing about genes and proteins - see lines 19, 291, 379-80 943, 950 etc - e.g., proteins have C-termini, but genes do not, an ORF is not fused to a protein. Parts of the text relating to this need to be rewritten.

Line 15 - suggest adding "Incl2" here.

Lines 21, 33 - I don't think that "modulate" is the correct word to use here and the "natures" of the recipient bacteria are not changed, but different strains act as recipients?

Line 35, 62, 139 etc - what is meant by "unstable"? - there should be a better word.

Line 48 - "electron transfer"

Line 63 - suggest "that allows donor bacterial to transfer genetic material to a..."

Line 69 - is "activation" the best word here?

Line 91 - suggest "a bacterium can encode more than one pilus type" if that is what is meant?

Line 103 - suggest "diverse and subunits are larger proteins", if this is what is meant?

Line 105 - should be "consisting of", but just "acid, either methionine..." here would be better.

Line 223 etc "in triplicate", but "triplicates" is correct elsewhere - e.g. line 947

Lines 212, 239 - please reword "shufflon mating experiments"

Line 218-9 - "plates were allowed to dry in a"

Lines 227, 951, 954 - "transfer/conjugation frequencies with"?

Line 256 - "scraping"

Line 273 - suggest "mice" here.

Line 322 - "formed" is the wrong word here.

Line 329 -the protein name is PilS.

Line 338 - PilS is the subunit? 185 amino acids is the size of PilS?

Line 379-80 - suggest "can vary" instead of "fluctuates", "amino acid", not "amino acids" here.

Line 385 - these trinucleotides would be better in uppercase.

Line 391 - "totalling"

Line 401 - could just be "by deleting rci"

Lines 408, 945 etc - "wild type" is not really needed after TP114.

Line 431 - this doesn't really fit/is not needed here.

Line 461 - "absence of either pillin"?

Line 453 - "on solid media"?

Lines 494-6 - problems with wording.

Lines 518-21 - confusing.

Line 528 - are these really "immobilized"? It is more that other possible shufflon segments have been removed?

Line 950 - transfer frequencies are for TP114 derivatives carrying these pilV variants.

Line 966 - "other" seems wrong here, delete "conditions" or replace with "media".

10) Supplementary Material

This is generally useful, but as Supplementary Material would not be edited by the journal prior to publication it needs some attention to make it easier to look at.

Tables are normally placed before figures and could be condensed (single spacing and/or smaller font).

Table S1 numbers in swine isolate descriptions are not explained and PEC15-20 has 1000.0.

Table S2 - "5'→3'" could be added in the column heading and removed from all primer sequences.

Table S3 could easily be fitted on a single page e.g. widen organism column. Also "lenght" needs fixing.

The Supplementary Figures are not easy to look at in the current format - each needs to be on a separate page, ideally with its

complete legend (e.g. by single

Reviewer #2 (Comments for the Author):

The manuscript «The type IV pilus of plasmid TP114 displays adhesins conferring conjugation 2 specificity and is important for DNA transfer in the mouse gut microbiota» by Allard et al. is a well-written documentation of the importance of type IV pili in bacterial conjugation in a natural habitat (the mouse gut). Most of my comments relate to the way the project is introduced and presented - and not to the actual results that I find interesting and well-discussed.

Major comments:

Abstract and introduction:

it does not become clear from abstract or introduction what the species range of TP114 is. I had to go to figure 5 and to a supplementary table to find out that the study was 'only' performed with a set of E.coli strains, suggesting that the plasmid might be specific to E.coli? Line 22 in the Abstract is thus somewhat misleading, where it says "with different recipient bacteria" - this actually should read "with different recipient E.coli strains"? To avoid the impression that we are talking about species specificity (instead of strain specificity)?

Introduction:

I would argue that the distinction between type II secretion systems and type IV pili is completely artificial, seeing that these systems have the exact same makeup in terms of structural/functional units - in Gram-negative bacteria. The difference is rather historical, and is influenced (in part) by the wrong assumption that protein complexes that do different things (e.g. protein secretion vs DNA secretion or uptake) cannot have the same setup. I feel that this could be somewhat better represented in the introduction, rather than just claiming that T4P share "some similarities with the bacterial type II secretion systems" (line 50). I'm reacting specifically to the "some similarities" part of this sentence...

At the same time, it is quite improbable that the T4P in Gram-negative and Gram-positive bacteria or Archaea are the same (as suggested in line 54-57). At least the membrane components must be rather different by definition. These are only discussed for Gram-negatives later in the introduction in more detail...

Seeing that later also other types of pili are mentioned (e.g. the F-pilus), it might be good to also give a very brief overview of the different types of "pili" and an explanation what makes a type IV pilus different from a type I, II, or III pilus (again, this is historical nomenclature that needs an explanation in my opinion...)

Figures:

A major concern for this manuscript is the excessive use of supplementary figures. I think that most of the data presented in these figures should be part of the main text and not hidden in the supplement, and certainly figure S3 should!

Supplementary Figure S2 is not acceptable. The alignment "fragment" shown there is not very meaningful - especially as some of the "conserved" features are - according to the alignment - not shared in all sequences. Is it really credible that *Pseudomonas* and *Vibrio* pilins do not have a cleavable signal? If it is, why is this not discussed anywhere? And how should the reader assess the quality of the overall alignment if only the first residues are shown? Last but not least: the bottom 5 sequences do have an GxxxxE motif that looks pretty much like the others - just shifted to the right. This makes me wonder how reliable the whole alignment really is? It would also be nice to give sequence identifiers (Uniprot?) in the alignment so that one can find the full-length sequences in a database. I do understand that this is later given in table S3 but this table is unnecessary if you show that complete alignment AND give the accession numbers there?

Minor comments:

Abstract:

Line 15: please give a (bacterial) species or species range for the plasmid TP114 in the abstract. Is it Enterobacteriaceae?

Line 19-20: "shufflon"? "tyrosine recombinase"? This is all very detailed and comes without explanation... (in the abstract... I am of course aware that this is later explained in the introduction). Maybe rephrase to make this more "digestible" for readers who are not deep into the theory and practice of conjugative plasmids? At least give a definition of "shufflon" here...

Importance:

Line 36: "explains how the remodeling [of] the PilV adhesin" - "of" missing.

Introduction:

Line 48-49: "T4P figure amongst the most widespread bacterial structures" is rather useless information, needs rephrasing, and is also somewhat wrong (the most widespread, at 100% spread, are things like ribosomes, membranes, etc...). Skip this half-sentence?

Line 65: where does the F-pilus come from, now? Needs introduction.

Line 68: as the causality does not come across in this sentence, maybe skip "thus"?

Line 80-81: some confusion here (probably on my part): are you sure the signal is cleaved in the cytoplasm, not the periplasm? And are type III secretion signals not reserved for type III secretion systems (name-wise)? If the signals mentioned here are distinct from the actual type III secretion signals of the type III systems (see e.g. [https://doi.org/10.1016/S0966-842X\(00\)01836-9](https://doi.org/10.1016/S0966-842X(00)01836-9)), then a differential discussion of this fact would help at this point.

Line 132: "thin pilus"? Why is it thin, compared to what, and is this classification (if it is a classification) meaningful? This was not introduced anywhere?

Line 132-133: "We examined the structure of the thin pilus in fluorescence microscopy using maleimide-conjugated fluorophores" - is this really "structure"? All you see is the approximate length, and obviously the overall presence (or absence) of the pili. For any structural detail, the resolution of the method is not suitable. Have you considered electron microscopy? In any case, I suggest to avoid the term "structure" in this context.

Line 133-135: multiple "also"

Line 139 should say "allows"?

Methods section:

Line 145: "broth medium"? I guess one of the two will do?

Line 186-187: you cannot "generate" a "predicted tertiary structure". What you generated is a model (of a structure).

Line 194 (and probably elsewhere): "liquid conjugation" sounds weird to me (lab slang?)... you mean conjugation in liquid medium...

Line 288-289: "...and protein homology between predicted open reading frames (ORF) was determined by BLASTp" - technically, BLAST can only determine sequence similarity, not homology. Note also that these sequences are homologous by definition as they very obviously have the same evolutionary origin (so nothing to 'determine', there).

Results and Figures:

Line 345-346: "The PilS topology"? You mean "The predicted PilS topology" maybe? At this point of the text, no experimental evidence has been shown for the topology of PilS.

Supplementary figure 5 has a title that I find confusing ("immobilization")?

Discussion:

Line 485: should say "we dissected the essential role of the T4P encoded by [the] IncI2 conjugative plasmid TP114" - missing "the".

Line 576-577: "and provide important insights for developing new therapeutics to fight bacterial infections." I do appreciate the attempt to demonstrate the importance of this study, but please either give a concrete suggestion on how the interesting details on shufflases and mating pair interactions presented in this study might lead to new therapeutics, or alternatively delete this sentence. I'm really not seeing the therapeutic approach here, especially as the bacteria can clearly shuffle specificities around on short time scales.

Staff Comments:

Preparing Revision Guidelines

Please return the manuscript within 60 days; if you cannot complete the modification within this time period, please contact me. If you do not wish to modify the manuscript and prefer to submit it to another journal, please notify me of your decision immediately so that the manuscript may be formally withdrawn from consideration by Microbiology Spectrum.

Response to reviewers:

We are grateful to the reviewers for their detailed constructive comments. Our response to their evaluations can be found below in blue. Note that all referred lines below correspond to the file named 'Marked-up Manuscript', which also contains figures to facilitate the revision of the modifications.

Response to Reviewer #1:

1) TP114, Fig. 1.

The origin of TP114 is not clear. It seems to be a historic IncI2 plasmid? Brief details of this should be included and, if possible, a reference cited.

TP114 has been isolated from an *E. coli* strain in Scotland. This information has been added in the introduction (line 131) with a reference.

It is not clear what "DSM-4246 (DSMZ)" in Table S1 refers to - this needs to be explained.

The DSM-4246 (DSMZ) refers to the Leibniz Institute DSMZ which is a German collection of microorganisms and cell cultures (<https://www.dsmz.de/collection/catalogue/details/culture/DSM-4246>).

Also, what is the relevance of the behaviour of this plasmid? e.g. how does it relate to other Inc2 plasmid carrying important resistance genes?

IncI2 plasmids first gained attention because they carried various beta-lactamase genes. Other IncI2 plasmids also include a shufflon that could help the dissemination of those plasmids. TP114 is particularly interesting since its transfer rates in the mouse gut microbiota reach high levels (close to 100%). We also used this plasmid to deliver CRISPR-Cas9 using engineered probiotics that can eliminate >99.9% of target antibiotic-resistant *E. coli* in the gastrointestinal tract and treat a *Citrobacter rodentium* infection in a mouse model.

In Fig. 1 genes in yellow ("selection") include "aph-III", which I think should be aph(3')-I (from a search of GenBank with the protein sequence).

We thank the reviewer to point this out. We have corrected the name of the kanamycin resistance gene in Fig. 1.

Presumably the kanamycin resistance conferred by this gene is what was used to select for transconjugants carrying TP114, but this doesn't seem to be clearly stated anywhere. This resistance gene needs to be made more obvious in Fig. 1 and reference to kanamycin resistance needs to be added to Table S1 and something added to the text to explain the selection.

The kanamycin resistance gene of TP114 has been added in table S1 and Fig. 1 title. The antibiotics used for selection after conjugation have been specified at lines 503, 519-521, and 544.

It is not clear what other genes in yellow labelled e.g., "yadA", "077", "079", "081" have to do with "selection", e.g., "077" is listed as "unknown function" in Supplementary Data 1 of Neil et al. Please rethink the colours used for these genes.

We have revisited all genes associated with the yellow label and distributed them into two new groups: transposase (orange) that include 054, *yadA*, *yadB*, *tnpA*, 078, 079, and 080 or other functions (black) that include *kikA*, *yeaA*, 043, *ycgB*, 049, *hicB*, *yajC*, *ydhA*, and 089.

2) Inc I numbers

The "1" of IncI1 and the "2" of IncI2 are written as subscripts throughout this manuscript, which is not conventional - see PDFs of Refs 69, 70, 72, 73, etc. This needs to be corrected, including in figures and tables, relevant reference titles and supplementary material.

We thank the reviewer to point this out. We have corrected these subscripts in all documents and Figures.

3) Organisation etc.

The length of both the Introduction and Discussion should be reduced, by removing detail that is less relevant to this manuscript and reducing repetition with the Results.

We shortened the introduction section from 99 to 90 lines. We also made changes in the discussion section passing from 93 to 90 lines. Although this is not a major difference, we thought that it was enough since 5000 words are allowed for research articles in *Microbiology Spectrum* and we have only 4316 words.

Over 100 references is excessive for a non-review paper and these need to be rationalised. For example, lines 42-48 include a large number of references - could a couple of reviews be used instead? - and the information in lines 49-57 is not that relevant to the remainder of the manuscript and cites a lot of references.

References have been revised and only those that were considered more important were kept, especially in the mentioned sections (went from 107 references to 66).

Lines 296-310 - it may be possible to reduce the detail needed here by referring to manufacturers' instructions and/or previous papers.

This is a custom method developed specifically to investigate the shufflon arrangement, hence making the use of references or manufacturer's instructions impossible. However, we have reviewed the paragraph and shortened the description as much as possible without losing key information.

Also, some long blocks of text could be divided into paragraphs e.g. new paragraph at the end of line 469.

We split this long paragraph into two shorter ones.

4) Descriptions of shufflons Lines 112-129, 397

Descriptions of shufflons and how they work are not clear/accurate - suggest something like "that can rearrange, changing the end..." on line 112, "rearrangement of shufflon segments, switching the 3' end of pilV" on line 120 (references are also needed here) and "switch changes the C-terminal end of PilV" on line 126, but repetition in this section also needs to be reduced

The description of the shufflon was modified (see lines 117-122 and 199-201) to reduce repetitions and clarify the description.

Lines 397, 408 etc - suggest something simpler, such as "different shufflon arrangements" rather than "conformational heterogeneity".

The wording 'conformational heterogeneity' has been replaced by 'shufflon arrangements', now at lines 218 as proposed.

Line 420 - suggest replacing with something like "pilV genes encoding PilV variants that..."

This modification has been done now at line 242.

Line 504-5 - is "controlled" the right word here and is there evidence that this is random?

The word 'randomly' has been removed at line 326 since Brouwer et al. 2019 stated that the recombination that occurs within the shufflon is not random and sometimes appeared to be host dependent.

5) Conjugation experiments, Lines 148-155, 195, 209-224

A lot of the Methods section is taken up with descriptions of different conjugation experiments. There seems to be some repetition and it is not clear why there are all these different variations, what would be used for selection for each experiments and the relevance of DAP.

The description of conjugation experiments has been reduced by referring to a previous manuscript (Neil et al. 2020). The selection of donors, recipients and exconjugants have been added (see lines 503, 519-521, and 544). Since recipient bacteria used in high-throughput conjugation assays had different antibiotic resistance profiles, the use of a $\Delta dapA$ donor strain considerably simplifies the procedure. Indeed, the $\Delta dapA$ donor strain cannot grow without diaminopimelic acid supplementation, leaving only the recipient or transconjugant bacteria for quantification after the matrin assay.

Lines 134, 136 - "host range" may suggest different species, but all recipients used here seem to be *E. coli*? Only *E. coli* are listed in Table S1, so what is this "broad range of Enterobacteriaceae"?

This paragraph was modified and the issue about host range was corrected (see line 135). This point was also clarified in the abstract (line 30)

Line 222 - cells were washed off plates into PBS? What about the liquid cultures?

We modified this section and now refer to a published article (Neil et al. 2020) describing the details of the method (line 469). To answer the question about liquid cultures, PBS was also added to the mating mixture in order to have the same final volume as matings performed on plates prior to doing serial dilutions.

Line 234 - what were the "appropriate antibiotics" for selecting. No antibiotic resistance is listed for most strains in Table S1/Fig. 5 and these must be added to understand how these recipients were selected for.

Line 269 - how was conjugation monitored? What selection was used?

We now referred to Neil *et al.* 2020 for the specific way conjugation in mice was monitored and the antibiotic selection(s) have now been added (lines 503, 519-521, and 544).

5) Homology, line 180, line 288, Fig. 3, line 939

Homology cannot be quantified, so it is incorrect to use "the percentage of homology". % identity must be used for nucleotide sequences while % identity or % similarity can be used when comparing protein sequences, depending on what has been calculated.

The term 'homology' has been changed for 'percent of protein identity' (lines 538, 803 and 806).

Line 180 - are these 40 bp regions actually identical?

Indeed, for the recombineering protocol, the 40 bp regions are identical to the corresponding locus to allow homologous recombination between the PCR amplified resistance gene and the plasmid/genome sequence to be modified. We have changed 'homology regions' for 'homology extensions' (line 435) which is the term used in Datsenko and Wanner, 2000. PNAS.

6) Lines 325-416

How does this relate to what is already known for other IncI2 plasmids such as R721? There are papers on the shufflon of R721. What new information is provided here? This is not really clear.

The shufflon of R721 has been described elsewhere (Komano *et al.* 1990, Plasmid; Kim and Komano 1992, J. Bacteriol). However, the description of PilS and especially the visualization of a T4P implicated in bacterial conjugation by fluorescent microscopy is novel.

Lines 21, 395 - searches with this segment suggest that it is present in many other plasmid sequences in GenBank - do you mean not yet described in a paper? This needs to be explained more accurately.

Indeed, we noticed that fragment D in TP114 is also present in other plasmids, but its official description in a shufflon has not been published yet. We clarified this point at line 216.

Lines 326-8 - it would seem more relevant to compare with another IncI2 plasmid here, rather than IncI1 R64, as the two types are not that closely related. Also line 390.

We compared with R64 instead of IncI2 plasmids since the organization of the *pil* genes is identical for IncI2 plasmids (TP114, R721, pChi7122-3). At line 390 (now 212) we compared TP114 to both R64 and R721 because TP114's shufflon share ORFs with both, especially a segment present in R64, but not in R721.

Lines 334-5- why is this interesting? Isn't this to be expected and TP114 PilS is closer to PilS R721, the other IncI2 plasmid, than to R64.

Indeed, it was expected that major pilins of TP114 and R64 would cluster together. We change this sentence now at line 151.

Line 346 -how is this known?

When comparing the predicted topology of PilS from TP114 (Supplementary figure S3A) with the one of R64 (Shimoda *et al.* 2008, J. Bacteriol) they are highly similar. We added information in the manuscript for that (lines 163-164).

Line 349- this needs a reference.

We have added a reference for this statement (line 168).

Line 352 - this needs more explanation.

The sentence has been split into two to clarify the way cysteine knock-in mutations have been introduced in TP114 and eB-TP114 (lines 170-173).

Lines 392, 543 - the "B" segments of R64 and TP114 are not that closely related - 55% identity according to Fig. 3, which is not that high, and line 543 says >75% and the D ORF is not related.

Indeed, the B segment of R64 and TP114 share 55% identity (45/82) but also 73% (60/82) positives i.e. amino acids having the same properties. We consider that this is sufficiently similar to state that those proteins are homologues. The section about the D ORF was removed from the text.

7) Other scientific points

Lines 77-8 - "PilS/A/E" and "PilV/E/W/X" need some explanation i.e. what that these subunits have different names in different systems.

Although these subunits are homologs, they don't have the same names because they were named by different research groups. Jacobsen *et al.* 2020, *Med. Microbiol. Immunol* stated: Although the T4P assembly system is conserved between bacterial species, the nomenclature of T4P proteins is very heterogeneous. Melville and Craig 2013, *Mircobiol. Mol. Biol. Rev.* also stated: Unfortunately, the nomenclature for homologous protein varies from species to species.

Lines 94, 326 - genes are e.g. carried, not encoded - they encode proteins. Also, lines 377, 502 - the shufflon is not "encoded" and "encoded for" is like saying "coded for for".

The term 'encode' have been replaced by 'comprise' at line 94 (now 96) and 326 (now 143) or by 'contain' at line 377 (now 198) and 502 (now 324)

Line 116 - the term "I complex" could be introduced here.

We now define what is the I-complex at lines 115-116 as suggested.

Lines 236, 244 - why 1.0% arabinose in one place and 0.1% in another? To only induce a low level of Rci?

As specified at lines 410, high induction of the shufflase appears to have a negative impact on strain fitness, so we therefore used 0.1% arabinose for this specific experiment. Otherwise, P_{BAD} induction was done with 1.0% arabinose.

Line 287 - should this be MF521836. 2? It would be good if this GenBank entry included a note explaining that the order of shufflon segments is only one possible (the dominant?) arrangement- see e.g., GenBank entries for R64, R721.

We thank the reviewer to point this out, the '.2' have now been added (line 536).

Lines 290-92 - "Sanger trace/chromatogram"? It is not clear why this was done?

We changed the word 'spectrum' for 'chromatogram' at line 539 and we referred to Supplementary Figure S4A at line 544 because that analysis was used to generate this Figure.

Lines 405-6 - this is not clear. Illumina sequencing was performed on extractions of whole genomic DNA and then the 39 bp signatures were identified in the raw reads? This needs to be rewritten.

We change the wording used to explain those results at lines 225-229.

8) Figures

Fig. 1 - see comments above

Fig. 2A is not that relevant to the main manuscript and seems to show the same information as Fig. S3A.

Indeed, Fig.2A reports information related to Fig. S3A. However, we think that Fig. 2A is still relevant in the main manuscript to show the predicted 3D structure of PilS and the location of the cysteine knock-ins. Since reviewer #2 suggested putting Fig. S3A in the main manuscript, which is in contradiction with the present request, we decided to let the figure contents as is.

Fig. 3 - the accession numbers for R64 and R721 should be added. See comments on homology above.

The GenBank accession numbers of R64 and R721 have been added at lines 537 and 802 (legend of Fig. 3).

Fig. 5 - numbers of strains, rather than %, should really be used here, as the total is below 100 (also line 447).

The number of strains instead of the % has been indicated in Fig. 5 and appropriate changes have been made at line 447 (now 268) and in the figure legend.

Fig 6 legend - the amount of experimental detail can be reduced - this should all be in Methods or Results.

Unnecessary details in the legend of Fig. 6 have been removed since they were already in the Methods section.

9) Minor points/English etc

All those suggestions have been taken into account and appropriate changes have been made at the line number specified after the comment.

Throughout - care needs to be taken when writing about genes and proteins - see lines 19, 291, 379-80 943, 950 etc - e.g., proteins have C-termini, but genes do not, an ORF is not fused to a protein. Parts of the text relating to this need to be rewritten.

Done

Line 15 - suggest adding "Incl2" here.

Done

Lines 21, 33 - I don't think that "modulate" is the correct word to use here and the "natures" of the recipient bacteria are not changed, but different strains act as recipients?

We are now using "allow conjugative transfer" instead of "modulate conjugative transfer" (lines 29). We also indicate that different *E. coli* strains were used as recipient bacteria.

Line 35, 62, 139 etc what is meant by "unstable"? - there should be a better word.

By unstable environment, we meant places where bacterial mobility, flow forces, and other environmental factors could perturb the interaction between the donor and recipient cells i.e. the intestinal gut. We have reworded this expression in the abstract and importance section and give a definition at line 62 (now 66-67).

Line 48 - "electron transfer"

This sentence has been removed to reduce the length of the Introduction section.

Line 63 – suggest "that allows donor bacterial to transfer genetic material to a..."

We have reworded this sentence to improve clarity (line 67).

Line 69 - is "activation" the best word here?

We now use 'deployment' of the T4SS (line 74)

Line 91 - suggest "a bacterium can encode more than one pilus type" if that is what is meant?

We have removed this sentence to shorten the introduction.

Line 103 – suggest "diverse and subunits are larger proteins", if this is what is meant?

We have reworded this sentence to improve clarity (line 104)

Line 105 - should be "consisting of", but just "acid, either methionine..." here would be better.

We have slightly modified this sentence to improve readability (line 104-107)

Line 223 etc "in triplicate", but "triplicates" is correct elsewhere - e.g. line 947

We have modified the text accordingly (see line 814).

Lines 212, 239 - please reword "shufflon mating experiments"

We apologize if this expression is not appreciated, but we did not find a better way to formulate it.

Line 218-9 - "plates were allowed to dry in a"

This section has been removed and the reader is now referred to Neil *et al.* 2020 for details.

Lines 227, 951, 954 – "transfer/conjugation frequencies with"?

We are not completely sure to understand the requested change here. We have reviewed the sentence to ensure that they were appropriately conveying the desired information.

Line 256 - "scraping"

The correction was made (line 503)

Line 273 – suggest “mice” here.

Done (line 522)

Line 322 - "formed" is the wrong word here.

We have replace “formed” with “separate them” (line 149)

Line 329 -the protein name is PilS.

Done. All gene and protein names were carefully reviewed.

Line 338 - PilS is the subunit? 185 amino acids is the size of PilS?

We changed ‘subunit’ for ‘protein’ (line 155)

Line 379-80 - suggest "can vary" instead of "fluctuates", "amino acid" , not "amino acids" here.

Done (line 199-201)

Line 385 – these trinucleotides would be better in uppercase.

Done (lines 205-206)

Line 391 - "totalling"

Done (line 211)

Line 401 – could just be “by deleting rci”

We have kept “by deleting the rci gene” to avoid any confusion (line 222)

Lines 408, 945 etc - "wild type" is not really needed after TP114.

Done (lines 224, 812)

Line 431 – this doesn’t really fit/is not needed here.

The sentence was removed (line 253)

Line 461 - "absence of either pillin"?

Done (line 283)

Line 453 – “on solid media”?

This was replace by “on solid medium” (line 274)

Lines 494-6 - problems with wording.

This sentence was reworded (line 314-316)

Line 508 – suggest “more than seven different partial ORFS”

Done (line 330)

Lines 518-21 - confusing.

This sentence was reworded to improve clarity (lines 338-341).

Line 528 - are these really "immobilized"? It is more that other possible shufflon segments have been removed?

Yes, the variants are really immobilized (deletion of the rci gene and all other partial ORF of the shufflon). We now refer the reader to Supplementary Figure S5B showing TP114 derivatives made to immobilize each variant (line 349).

Line 950 – transfer frequencies are for TP114 derivatives carrying these pilV variants.
(line 817)

Yes, the sentence was modified to avoid any confusion.

Line 966 - "other" seems wrong here, delete "conditions" or replace with "media".

We only use the term “condition” (lines 832-833)

10) Supplementary Material

This is generally useful, but as Supplementary Material would not be edited by the journal prior to publication it needs some attention to make it easier to look at. Tables are normally placed before figures and could be condensed (single spacing and/or smaller font).

This has been changed in the updated version of the document.

Table S1 numbers in swine isolate descriptions are not explained and PEC15-20 has 1000.0.

The appropriate corrections have been made (Table S1). The detailed description of the strains were not available in the cited article, and we have not yet been able to obtain this information from the corresponding author.

Table S2 - "5'→3'" could be added in the column heading and removed from all primer sequences.

This has been changed in the updated document.

Table S3 could easily be fitted on a single page e.g. widen organism column. Also "lenght" needs fixing.

Table S3 was removed according to the comment made by reviewer #2. The information is now presented in Supplementary Figure S2.

The Supplementary Figures are not easy to look at in the current format - each needs to be on a separate page, ideally with its complete legend

This has been fixed in the updated document.

Response to Reviewer #2:

The manuscript «The type IV pilus of plasmid TP114 displays adhesins conferring conjugation 2 specificity and is important for DNA transfer in the mouse gut microbiota» by Allard et al. is a well-written documentation of the importance of type IV pili in bacterial conjugation in a natural habitat (the mouse gut). Most of my comments relate to the way the project is introduced and presented - and not to the actual results that I find interesting and well-discussed.

Major comments:

Abstract and introduction:

it does not become clear from abstract or introduction what the species range of TP114 is. I had to go to figure 5 and to a supplementary table to find out that the study was 'only' performed with a set of E.coli strains, suggesting that the plasmid might be specific to E.coli? Line 22 in the Abstract is thus somewhat misleading, where it says "with different recipient bacteria" - this actually should read "with different recipient E.coli strains"? To avoid the impression that we are talking about species specificity (instead of strain specificity)?

This issue has been corrected to avoid any ambiguity for the readers.

Introduction:

I would argue that the distinction between type II secretion systems and type IV pili is completely artificial, seeing that these systems have the exact same makeup in terms of structural/functional units - in Gram-negative bacteria. The difference is rather historical, and is influenced (in part) by the wrong assumption that protein complexes that do different things (e.g. protein secretion vs DNA secretion or uptake) cannot have the same setup. I feel that this could be somewhat better represented in the introduction, rather than just claiming that T4P share "some similarities with the bacterial type II secretion systems" (line 50). I'm reacting specifically to the "some similarities" part of this sentence...

We have changed 'some similarities' and develop this point by writing: 'and share core homologous components and similarities in terms of macromolecular architecture'.

At the same time, it is quite improbable that the T4P in Gram-negative and Gram-positive bacteria or Archaea are the same (as suggested in line 54-57). At least the membrane components must be rather different by definition. These are only discussed for Gram-negatives later in the introduction in more detail...

We added 'Despite some structural differences' at the beginning of the sentence to highlight that those T4P are not identical in those different bacteria (line 60-61).

Seeing that later also other types of pili are mentioned (e.g. the F-pilus), it might be good to also give a very brief overview of the different types of "pili" and an explanation what makes a type IV pilus different from a type I, II, or III pilus (again, this is historical nomenclature that needs an explanation in my opinion...)

We understand that the terminology of pili is not known by everyone. However, it seems impossible to briefly summarize this historical nomenclature. A full paragraph should be added to address this topic. Since Reviewer #1 already asked us to reduce the length of the introduction, we decided not to present these details. Paranchych and Frost, 1988, *Adv. Microb. Physiol.* explains the discovery of pilus and the associated nomenclature.

Figures:

A major concern for this manuscript is the excessive use of supplementary figures. I think that most of the data presented in these figures should be part of the main text and not hidden in the supplement, and certainly figure S3 should!

We understand this concern, however, the supplementary figures are not 'hidden' and are easily accessible online. We think that 6 figures in the main text are more than enough. Moreover, the information contained in Supplementary Figures is interesting, but not essential to the understanding of the manuscript. As for figure S3 (which complements the information of Fig. 2), reviewer # 1 believed that figure 2A should be removed from the main text. Since this is in contradiction with the present request, we decided to let the figure contents as is.

Supplementary Figure S2 is not acceptable. The alignment "fragment" shown there is not very meaningful - especially as some of the "conserved" features are - according to the alignment - not shared in all sequences. Is it really credible that *Pseudomonas* and *Vibrio* pilins do not have a cleavable signal? If it is, why is this not discussed anywhere? And how should the reader assess the quality of the overall alignment if only the first residues are shown? Last but not least: the bottom 5 sequences do have an GxxxxE motif that looks pretty much like the others - just shifted to the right. This makes me wonder how reliable the whole alignment really is? It would also be nice to give sequence identifiers (Uniprot?) in the alignment so that one can find the full-length sequences in a database. I do understand that this is later given in table S3 but this table is unnecessary if you show that complete alignment AND give the accession numbers there?

We thank the reviewer to point this out. Indeed, since pilin sequences that are part of the type IVc group are shorter, the cleavage site is shifted to the right compared to the other sequences. We have added the predicted cleavage site for this group and indicated the glutamate at position 5 in a box. We initially decided to show only the beginning of the alignment since the main features that we highlighted were found in this section of the proteins. However, we have prepared a figure with the whole alignment as requested.

Minor comments:

Abstract:

Line 15: please give a (bacterial) species or species range for the plasmid TP114 in the abstract. Is it Enterobacteriaceae?

We have added the term 'enterobacterial' to point out the species range of the plasmid TP114 (now line 30).

Line 19-20: "shufflon"? "tyrosine recombinase"? This is all very detailed and comes without explanation... (in the abstract... I am of course aware that this is later explained in the introduction). Maybe rephrase to make this more "digestible" for readers who are not deep into the theory and practice of conjugative plasmids? At least give a definition of "shufflon" here...

We have added the definition of a shufflon and given more details about the tyrosine recombinase (now lines 23-28).

Importance:

Line 36: "explains how the remodeling [of] the PilV adhesin" - "of" missing.

We are sorry but we don't see where the "of" is missing in this sentence.

Introduction:

Line 48-49: "T4P figure amongst the most widespread bacterial structures" is rather useless information, needs rephrasing, and is also somewhat wrong (the most widespread, at 100% spread, are things like ribosomes, membranes, etc...). Skip this half-sentence?

This part of the sentence has been removed.

Line 65: where does the F-pilus come from, now? Needs introduction.

The mating pair stabilization step can be done by T4P, but also by the conjugative F-pilus so we thought that it was important to mention it. We have now rewritten this sentence, line 70.

Line 68: as the causality does not come across in this sentence, maybe skip "thus"?

The word 'thus' has been removed now at line 73.

Line 80-81: some confusion here (probably on my part): are you sure the signal is cleaved in the cytoplasm, not the periplasm? And are type III secretion signals not reserved for type III secretion systems (name-wise)? If the signals mentioned here are distinct from the actual type III secretion signals of the type III systems (see e.g. [https://doi.org/10.1016/S0966-842X\(00\)01836-9](https://doi.org/10.1016/S0966-842X(00)01836-9)), then a differential discussion of this fact would help at this point.

Indeed, the Type III signal sequence is cleaved in the cytoplasm and not in the periplasm. Here we refer to Type III "Signal Sequence" (not secretion signal), which is only present in the

prepilins. 'Type III' is by opposition to the type I (recognized by signal peptidase I) and type II (characteristic of lipoproteins) signal sequences. (see Giltner *et al.* 2012, Microbiol. Mol. Biol. Rev. for more details).

Line 132: "thin pilus"? Why is it thin, compared to what, and is this classification (if it is a classification) meaningful? This was not introduced anywhere?

The term 'thin' is in opposition to 'thick' and their corresponding sizes are now defined at line 159.

Line 132-133: "We examined the structure of the thin pilus in fluorescence microscopy using maleimide-conjugated fluorophores" - is this really "structure"? All you see is the approximate length, and obviously the overall presence (or absence) of the pili. For any structural detail, the resolution of the method is not suitable. Have you considered electron microscopy? In any case, I suggest to avoid the term "structure" in this context.

Indeed, fluorescence microscopy doesn't offer a high enough resolution to see the actual "structure" of T4P. We have removed the term 'structure' (now at line 133).

Line 133-135: multiple "also"

This passage has already been reworded, which removed the accumulation of 'also' (now at lines 133-134).

Line 139 should say "allows"?

Thanks, we made this change, now line 138.

Methods section:

Line 145: "broth medium"? I guess one of the two will do?

We removed 'medium', see line 401.

Line 186-187: you cannot "generate" a "predicted tertiary structure". What you generated is a model (of a structure).

We have rewritten this sentence to be more accurate, see line 443.

Line 194 (and probably elsewhere): "liquid conjugation" sounds weird to me (lab slang?)... you mean conjugation in liquid medium...

We have changed all 'liquid conjugation' for 'conjugation in broth' in the entire manuscripts.

Line 288-289: "...and protein homology between predicted open reading frames (ORF) was determined by BLASTp" - technically, BLAST can only determine sequence similarity, `_not_`

homology. Note also that these sequences are homologous by definition as they very obviously have the same evolutionary origin (so nothing to 'determine', there).

We change this section of the text for 'the percent of protein identity was determined by BLASTp', now at line 538.

Results and Figures:

Line 345-346: "The PilS topology"? You mean "The predicted PilS topology" maybe? At this point of the text, no experimental evidence has been shown for the topology of PilS.

We added the word 'predicted', line 163.

Supplementary figure 5 has a title that I find confusing ("immobilization")?

The title has been changed for 'Representation of TP114 derivatives', line 83 of the supplemental materials document

Discussion:

Line 485: should say "we dissected the essential role of the T4P encoded by [the] IncI2 conjugative plasmid TP114" - missing "the".

Done, line 307.

Line 576-577: "and provide important insights for developing new therapeutics to fight bacterial infections." I do appreciate the attempt to demonstrate the importance of this study, but please either give a concrete suggestion on how the interesting details on shufflases and mating pair interactions presented in this study might lead to new therapeutics, or alternatively delete this sentence. I'm really not seeing the therapeutic approach here, especially as the bacteria can clearly shuffle specificities around on short time scales.

We have now rewritten the last two sentences of the discussion to clarify our thoughts (lines 394-396).

February 16, 2022

Prof. Sébastien Rodrigue
Université de Sherbrooke
Biologie
2500 Boulevard Université
Sherbrooke, Quebec J1K 2R1
Canada

Re: Spectrum02303-21R1 (The type IV pilus of plasmid TP114 displays adhesins conferring conjugation specificity and is important for DNA transfer in the mouse gut microbiota.)

Dear Prof. Sébastien Rodrigue:

Your manuscript has been accepted, and I am forwarding it to the ASM Journals Department for publication. You will be notified when your proofs are ready to be viewed.

Sincerely,

Jennifer Auchtung
Editor, Microbiology Spectrum
